# Intestinal Dominance by Multidrug-Resistant Bacteria in Pediatric Liver Transplant Patients

Elias Dahdouh,[a] Lorena Fernández-Tomé,[b] Emilio Cendejas-Bueno,[a] Guillermo Ruiz-Carrascoso,[a] Cristina Schüffelmann,[c] María Alós-Díez,[b] Fernando Lázaro-Perona,[a] Mercedes Castro-Martínez,[d] Luis Escosa-García,[e,f] Sonia Jiménez-Rodríguez,[a] Loreto Hierro-Llanillo,[b] Jesús Mingorance[a,f]

aClinical Microbiology and Parasitology Department, Hospital Universitario La Paz, IdiPAZ, Madrid, Spain
bPediatric Hepatology Department, Hospital Universitario La Paz, IdiPAZ, Madrid, Spain
cPediatric Intensive Care Unit, Hospital Universitario La Paz, Madrid, Spain
dPreventive Medicine Department, Hospital Universitario La Paz, Madrid, Spain
ePediatric Tropical and Infectious Diseases Department, Hospital Universitario La Paz, Madrid, Spain
fCIBERINFEC, Instituto de Salud Carlos III, Madrid, Spain

**ABSTRACT** Pediatric liver transplantation (PLTx) is commonly associated with extensive antibiotic treatments that can produce gut microbiome alterations and open the way to dominance by multidrug-resistant organisms (MDROs). In this study, the relationship between intestinal Relative Loads (RLs) of $\beta$-lactamase genes, antibiotic consumption, microbiome disruption, and the extraintestinal dissemination of MDROs among PLTx patients is investigated. 28 PLTx patients were included, from whom 169 rectal swabs were collected. Total DNA was extracted and $bla_{CTX-M-1-Family}$, $bla_{OXA-1}$, $bla_{OXA-48}$, and $bla_{VIM}$ were quantified via quantitative polymerase chain reaction (qPCR) and normalized to the total bacterial load (*16SrRNA*) through Log$\Delta\Delta$Ct to determine the RLs. *16SrRNA* sequencing was performed for 18 samples, and metagenomic sequencing was performed for 2. Patients' clinical data were retrieved from the hospital's database. At least one of the genes tested were detected in all of the patients. The RLs for $bla_{CTX-M-1-Family}$, $bla_{OXA-1}$, $bla_{OXA-48}$, and $bla_{VIM}$ were higher than 1% of the total bacterial population in 67 (80.73%), 56 (78.87%), 57 (77.03%) and 39 (61.9%) samples, respectively. High RLs for $bla_{CTX-M-1-Family}$, $bla_{OXA-1}$, and/or $bla_{OXA-48}$, were positively associated with the consumption of carbapenems with trimethoprim-sulfamethoxazole and coincided with low diversity in the gut microbiome. Low RLs were associated with the consumption of noncarbapenem $\beta$-lactams with aminoglycosides ($P < 0.05$). Extraintestinal isolates harboring the same gene(s) as those detected intraintestinally were found in 18 samples, and the RLs of the respective swabs were high. We demonstrated a relationship between the consumption of carbapenems with trimethoprim-sulfamethoxazole, intestinal dominance by MDROs and extraintestinal spread of these organisms among PLTx patients.

**IMPORTANCE** In this study, we track the relative intestinal loads of antibiotic resistance genes among pediatric liver transplant patients and determine the relationship between this load, antibiotic consumption, and infections caused by antibiotic-resistant organisms. We demonstrate that the consumption of broad spectrum antibiotics increase this load and decrease the gut microbial diversity among these patients. Moreover, the high loads of resistance genes were related to the extraintestinal spread of multidrug-resistant organisms. Together, our data show that the tracking of the relative intestinal loads of antibiotic resistance genes can be used as a biomarker that has the potential to stop the extraintestinal spread of antibiotic-resistant bacteria via the measurement of the intestinal dominance of these organisms, thereby allowing for the application of preventive measures.

Address correspondence to Elias Dahdouh, elie.dahdouh@gmail.com.

The authors declare no conflict of interest.

*[This article was published on 8 November 2022 with Emilio Cendejas-Bueno listed as Emilio Cendejas and Sonia Jiménez-Rodríguez listed as Sonia Jiménez-Díaz. The author names were updated in the current version, posted on 15 November 2022.]*

**KEYWORDS** intestinal dominance, pediatric liver transplant patients, $\beta$-lactamase genes, relative intestinal load, qPCR, antibiotic consumption, extraintestinal multidrug-resistant organisms, microbiome

Liver transplantation is a life-saving procedure for pediatric patients suffering from deadly conditions that are otherwise untreatable (1). Due to the great advances in the past decades, the 5-year survival rate of these patients is above 85% (2). Nevertheless, pediatric liver transplant patients remain rather difficult to manage, due to a set of challenges that are less frequently encountered in other types of solid organ transplantations (3). One of these challenges is managing the high risk for perioperative infections that are caused by the highly invasive procedure performed, especially since the transplanted organ is directly connected to the intestine and is accessible to the gut microbiome (4). Another challenge is managing the late-onset infections that are usually the result of the hard to achieve equilibrium between the adequate level of immunosuppression that prevents organ rejection and the level of oversuppression that gives rise to opportunistic infections (5). Moreover, the start of immunosuppressive therapies at times when children's immune systems are still under development leads to higher rates of infections, compared to adult liver transplant patients (6, 7). These infections, together with malignancies and immunosuppressive complications, are responsible for two thirds of the late deaths after transplant (5) and are associated with high rates of morbidity and mortality (6).

Due to the high risk of bacterial infections, pediatric liver transplantation protocols require the administration of high doses of prophylactic antibiotics over long periods of time. Moreover, broad-spectrum antibiotics are frequently administered in response to infections (8). Though this extensive use of antibiotics can be beneficial for some patients, it is not without consequences. One such consequence is driving bacterial resistance that results in high rates of infections with multidrug-resistant organisms (MDROs) that produce extendeds-spectrum $\beta$-lactamases (ESBLs) and/or carbapenemases (9). These infections are difficult to treat, are life-threatening, and can lead to severe sepsis (10). The CTX-M-1-Family is one of the most commonly detected ESBLs both worldwide and at the hospital in which this study was performed (11). Its gene ($bla_{CTX-M-1-Family}$) is commonly found on the same plasmid harboring $bla_{OXA-1}$, which does not produce an ESBL by itself but rather increases the activity of the CTX-M-1-Family ESBLs against penicillin/inhibitor combinations (12). In terms of resistance to carbapenems, the $bla_{OXA-48}$ and $bla_{VIM}$ carbapenemase genes are also present on plasmids that often contain other genes of antibiotic resistance. These genes have a wide global dissemination, result in high rates of resistance to carbapenems, and are endemic at our hospital (13, 14).

Another consequence of the extensive use of antibiotics is the disruption of the intestinal microbiome, which has an important role in child development (15). This disruption could lead to intestinal dominance by MDROs (defined in this work as an increase in the abundance of these organisms to over 10% of the intestinal microbiome) (14, 16, 17), which, in turn, is linked to invasive infections (18) and horizontal transmission (19). In this study, we aim at tracking the extent of intestinal dominance by MDROs harboring the $bla_{CTX-M-1-Family}$, $bla_{OXA-1}$, $bla_{OXA-48}$, and/or $bla_{VIM}$ among pediatric transplant patients over time, in relation to the antibiotic treatments received, and in relation to the extraintestinal spread of these organisms.

## RESULTS

**Patient data and samples collected.** A total of 28 pediatric liver transplant patients (labeled P1 to P28) were included in this study from October of 2018 until December of 2019. 17 patients were included immediately after the transplant surgery (group 1), and 11 patients that had been transplanted before the study period were included during their follow-up period (group 2). For the group 2 patients, the transplant had been performed between 9 and 104 weeks before inclusion (median = 35 weeks). Table 1 shows the baseline clinical characteristics of the patients upon inclusion in the study.

**TABLE 1** Baseline clinical characteristics of the patients included in this study

| Sampling characteristic | Number of samples |
|---|---|
| No. of patients | 28 |
| Average age (yrs) | 4.1 (median = 1.6) |
| Sex | 15 females (53.57%) |
| Average weeks followed | 22.5 (median = 14.5; range = 1 to 60) |
| | |
| Problem Leading to Transplantation | |
| Biliary atresia | 12 (42.9%) |
| Hepatocarcinoma | 2 (7.1%) |
| Propionic acidemia | 2 (7.1%) |
| Maple syrup urine disease (MSUD) | 2 (7.1%) |
| Autoimmune hepatitis | 2 (7.1%) |
| Late graft failure (biliary) | 2 (7.1%) |
| Acute liver failure | 2 (7.1%) |
| Late graft failure (chronic rejection) | 1 (3.6%) |
| MDR3 deficiency (PFIC3) | 1 (3.6%) |
| Arginosuccinic aciduria | 1 (3.6%) |
| Classic Citrullinemia | 1 (3.6%) |

A total of 169 rectal swabs were collected from these patients for routine epidemiological screening and were included in this study. In 4 patients, 1 sample was obtained before transplant, and in another patient, 2 samples were obtained before transplant. All of the other samples were collected after the transplant surgery. Table 2 shows the characteristics of the samples obtained from these patients as well as the MDROs detected among these samples through routine cultures.

**Relative intestinal loads of antibiotic resistance genes in pediatric liver transplant patients.** The relative intestinal loads of $bla_{CTX-M-1-Family}$, $bla_{OXA-1}$, $bla_{OXA-48}$, and $bla_{VIM}$ were determined using the $Log\Delta\Delta C_t$ method (14, 20) and were expressed in terms of the percentage of the total bacterial population (%RL). The $C_t$ values of the *16SrRNA* gene did not have a significant difference between the different samples over the course of the study, reflecting no significant change in the total bacterial population (Slope = 0.023, $R^2$ = 0.0117). 83 (49.1%) rectal swabs were positive by quantitative polymerase chain reaction (qPCR) for $bla_{CTX-M-1-Family}$, 71 (42%) for $bla_{OXA-1}$, 74 (43.8%) for $bla_{OXA-48}$, and 63 (37.3%) for $bla_{VIM}$. The median %RLs for the positive samples were 4.2%, 5.1%, 5.6%, and 3.3%, respectively, for each of the tested genes.

Given that the total amount of *Enterobacterales* in a healthy gut microbiome is around 1% of the total bacterial population (21) and that the vast majority of MDROs identified in the rectal swabs are *Enterobacterales* (Table 2), the %RLs of these genes were divided as follows: Low RL for samples with %RL values below 1% ($Log\Delta\Delta C_t < -2$), High RL for samples with %RLs from 1% to 10% (i.e., within 1 order of magnitude; $Log\Delta\Delta C_t$ between $-2$ and $-1$), and Very High RL for samples with %RLs from 10% to 100% (i.e., within 2 orders of magnitude; $Log\Delta\Delta C_t$ between $-1$ and $0$). Fig. 1 shows the distribution of the RLs according to these designations as well as how the majority of the RLs in the positive samples were High or Very High.

Using qPCR resulted in detecting the ESBL gene $bla_{CTX-M-1-Family}$ in 65 samples, and the carbapenemase genes $bla_{OXA-48}$ and/or $bla_{VIM}$ were detected in 25 samples that were negative by routine, culture-based screening. The samples that were negative by selective culture media but positive by qPCR for $bla_{OXA-48}$ had a median %RL of 0.05% (range = 0.0005% to 16.64%) and were significantly lower than samples that were positive by both culture media and by qPCR (median = 6.35%; range = 0.0003% to 21.14%; $P < 0.05$). This was also observed for $bla_{VIM}$ (%RL of 0.38%, range = 0.0001% to 28.59% versus 6.35%, range = 0.0003% to 21.14%; $P < 0.05$). No significant difference was detected in the RLs for the ESBL selective media between the samples that were only positive by qPCR and those that were positive by both techniques.

**Changes in the relative intestinal loads of the resistance genes over time.** The change in RLs over time in 4 representative patients, together with the data of which

**TABLE 2** Characteristics of the rectal swabs collected and the multidrug-resistant isolates detected among them via routine epidemiological screening[a]

| Sampling characteristic | Number of samples |
| --- | --- |
| Total rectal swabs collected | 169 |
| Average swab per patient | 6 ± 4 (median = 5, range = 1 to 18) |
| | |
| Ward from which the rectal swabs were collected | |
| Hepatology (inpatient) | 113 (66.9%) |
| Pediatric Intensive Care Unit | 27 (16%) |
| Hepatology (outpatient) | 15 (8.9%) |
| Emergency | 10 (5.9%) |
| Pediatrics (outpatient) | 4 (2.4%) |
| | |
| Patients colonized by multidrug-resistant isolates | 22 (out of 28) |
| Carbapenemase producers | 13 |
| ESBL Producers | 4 |
| Carbapenemase and ESBL producers | 5 |
| | |
| Swabs in which multidrug organisms detected | 95 (56.2% of the total swabs collected) |
| *Klebsiella pneumoniae* | 74 (77.9%) |
| *Klebsiella aerogenes* | 7 (7.4%) |
| *Pseudomonas aeruginosa* | 6 (6.3%) |
| *Escherichia coli* | 5 (5.3%) |
| *Enterobacter cloacae* | 1 (1.1%) |
| *Citrobacter amalonaticus* | 1 (1.1%) |
| *Klebsiella oxytoca* | 1 (1.1%) |
| | |
| $\beta$-Lactamases detected among the multidrug-resistant organisms through routine screening | |
| Carbapenemases (unspecified) | 59 (62.1%) |
| ESBLs | 18 (18.9%) |
| VIM | 9 (9.5%) |
| OXA-48 | 9 (9.5%) |

[a]"ESBL" stands for extended spectrum beta-lactamase. "Carbapenemases (unspecified)" have been detected by phenotypic assays only through routine testing and were so registered in the hospital's information system database.

antibiotic(s) they were receiving at the time of sample collection, the episodes of extra-intestinal detection of MDROs, and the date of transplantation, are shown in Fig. 2. The graphs of the RLs over time for all 28 patients are shown in Supplementary File 1.

As per the protocol applied at the time of the study, all patients underwent intestinal cleansing with tobramycin and nystatin before transplantation. Vancomycin and piperacillin-tazobactam were then administered for 5 days (each), tobramycin for 21 days, and TMX for 6 months after surgery. The effect of prophylaxis was clear in 3 of the patients, as represented in Fig. 2 (P9, P10, and P20), in which the RLs of the 4 genes were negligible at the time of the surgery, as was the total bacterial population (i.e., high $C_t$ values for the *16SrRNA* gene). This drop was also observed in 10 patients from group 1. Nevertheless, in the 3 patients shown in Fig. 2, the RLs of the resistance genes quickly increased in the postoperative period, during which they received various courses of antibiotics and had stays at the intensive care unit. The exception was patient P5 (Fig. 2A), in whom the prophylaxis did not seem to lower the RLs, despite a drop in the total bacterial population.

In these patients, the RLs of the $\beta$-lactamase genes remained high when the patients were no longer receiving $\beta$-lactams and were only receiving aminoglycosides and/or trimethoprim-sulfamethoxazole (TMX). This also applies to seven patients from group 1 (26 out of 28 rectal swabs) and 8 patients from group 2 (16 out of 20 rectal swabs). Finally, all of the patients depicted in Fig. 2 had most of their extraintestinal episodes of MDRO infections when the RLs of the genes of resistance were around 10%.

**Antibiotic consumption and relative intestinal loads.** The antibiotic consumption data set from 30 days before the collection of each rectal swab was recovered from the

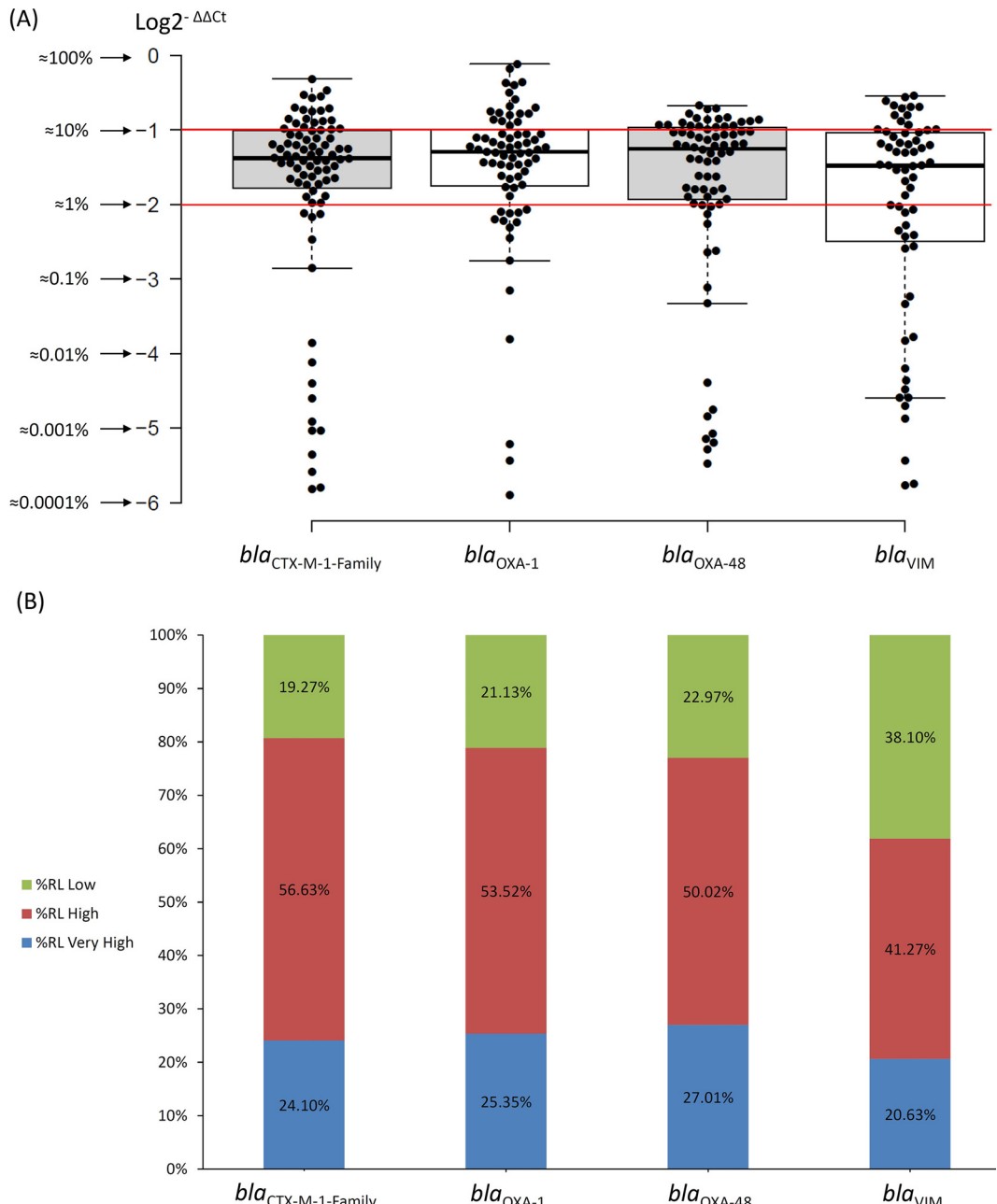

**FIG 1** Distribution of the % Relative Loads (%RLs) of the four genes tested for in this study according to the following designations. Low %RL, RLs below 1% of the total bacterial population; High %RL, RLs between 1% and 10% of the total bacterial population; Very High %RLs, RLs above 10% of the total bacterial population. The red lines in part (A) depict the division between the Very High, High, and Low RLs.

hospital's registries. In 2 patients, no antibiotics were used outside the prophylaxis protocol (described above) during the time they were included in the study. For the remaining patients, these and other antibiotics were administered when infections were suspected. Non-carbapenem $\beta$-lactams were administered to 22 patients in different combinations: alone in 5 patients, in combination with carbapenems ($n = 5$ patients), with carbapenems and aminoglycosides ($n = 5$), with carbapenems, aminoglycosides, and TMX ($n = 4$), with carbapenems and TMX ($n = 1$), with aminoglycosides and TMX ($n = 1$), and with TMX ($n = 1$). The most common noncarbapenem $\beta$-lactam used was piperacillin-tazobactam ($n = 9$), followed by ceftazidime-avibactam ($n = 7$), amoxicillin-clavulanic acid ($n = 6$), cefotaxime ($n = 3$), cefuroxime ($n = 2$), and ampicillin

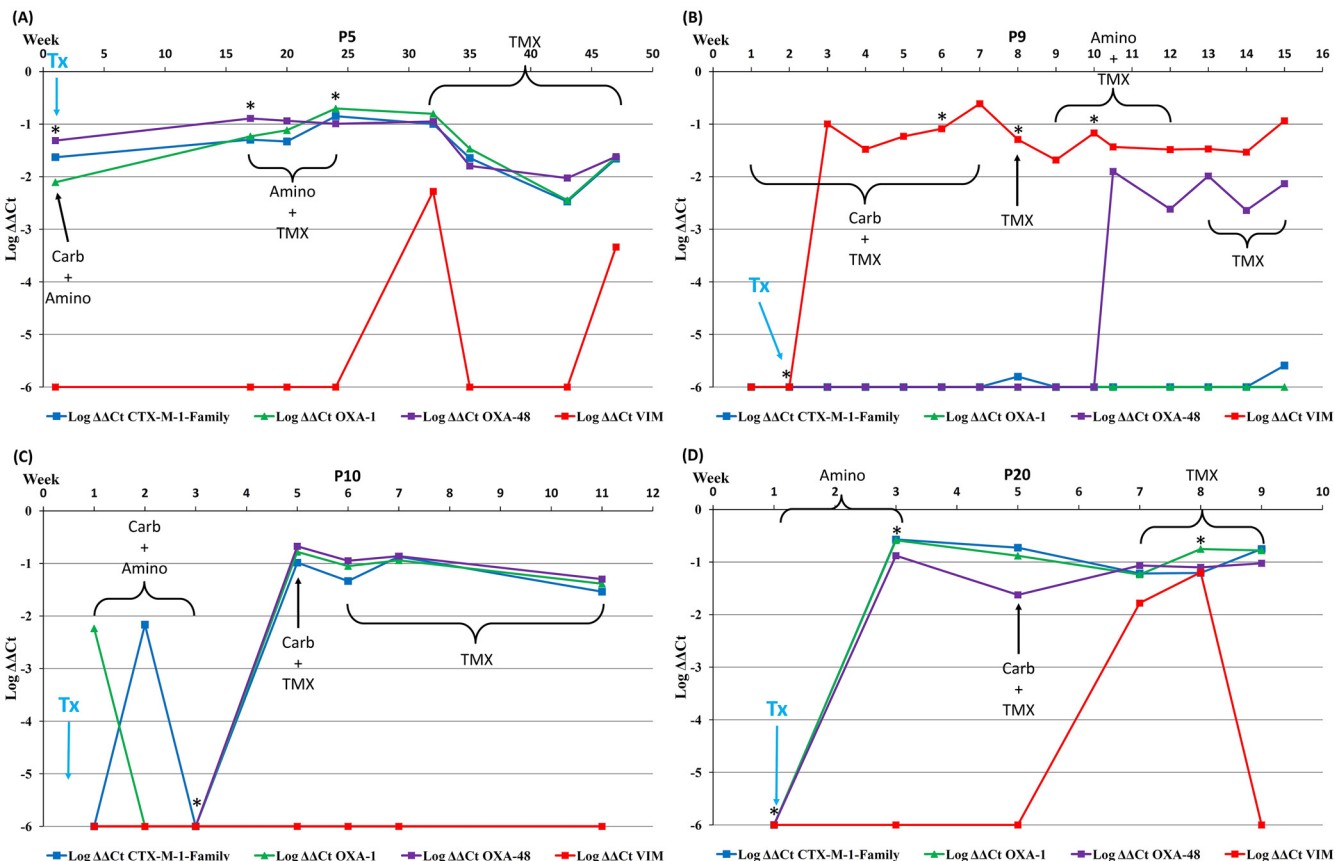

**FIG 2** Relative intestinal loads of $bla_{CTX-M-1-Family}$, $bla_{OXA-1}$, $bla_{OXA-48}$, and $bla_{VIM}$ for 4 patients combined with data regarding antibiotics received at the time of sample collection, the extraintestinal isolation of MDROs, and the date of transplantation. Carb, carbapenems; Amino, aminoglycosides; TMX, trimethoprim-sulfamethoxazole; Tx, date of transplantation; *, detection of extraintestinal MDROs. The patients were chosen as representatives of the most common patterns of the changes in intestinal loads of resistance genes over time. The change of the intestinal loads over time for all of the patients included in the study is found in Supplementary File 1.

($n = 1$). Several patients received more than one course of these antibiotics throughout the study period.

The antibiotics administered to both groups of patients were comparable in frequency, except for the aminoglycosides which were administered to 5 patients from group 2 and 14 patients of group 1. A significant association was determined between the consumption of noncarbapenem $\beta$-lactams and/or aminoglycosides during the previous 30 days and being negative for $bla_{CTX-M-1-Family}$, $bla_{OXA-1}$, and $bla_{OXA-48}$, compared to the consumption of carbapenems and TMX (Kruskal-Wallis test; $P < 0.05$). The consumption of carbapenems and TMX were positively associated with being positive for $bla_{CTX-M-1-Family}$, $bla_{OXA-1}$, and $bla_{OXA-48}$, as well as for having Very High or High RLs of these genes (Chi-square test; $P < 0.05$) (Table S1). The positive samples had more Very High and High RLs than Low RLs for all 4 genes, regardless of which antibiotic was administered (Table S2).

**Relative intestinal loads and extraintestinal spread of multidrug-resistant isolates.** In 9 out of the 28 patients included in this study, extraintestinal MDRO isolates were obtained within 14 days of collection of a rectal swab. 6 of these patients belonged to group 1, and 3 belonged to group 2. A total of 24 episodes were recorded, and in 19 of them, the rectal swab was obtained before the extraintestinal detection. Among the extraintestinal episodes, 19 were infections, as registered by the attending clinician. The diagnoses registered for these incidents, as well as whether the same species, gene, and clone were detected intraintestinally and extraintestinally, and the average RLs of the rectal swabs at the time of extraintestinal spread are shown in Table 3.

The RLs for $bla_{VIM}$ before the detection of extraintestinal MDRO isolates were significantly higher than the RLs with which there was no extraintestinal detection of MDROs

**TABLE 3** Detection of extraintestinal multidrug-resistant isolates in comparison with the data obtained from rectal swabs within 14 days of extraintestinal detection

| Sampling characteristic | Value |
| --- | --- |
| Extra-intestinal multidrug-resistant isolates | 24 |
| Cholangitis | 14 (58.3%) |
| Surgery-Related Peritonitis | 3 (12.5%) |
| Nosocomial Pneumonia | 1 (4.2%) |
| Catheter-Related Urinary Tract Infection | 1 (4.2%) |
| Asymptomatic Bacteruria | 4 (16.7%) |
| Pharyngeal Colonization | 1 (4.2%) |
| | |
| Same species detected intraintestinally and extraintestinally | 16 out of 19 (84.2%)[a] |
| Same gene detected intraintestinally and extraintestinally | 18 out of 23 (78.3%)[b] |
| Same clone detected intraintestinally and extraintestinally | 4 out of 4 (100%)[c] |
| | |
| Average RLs of the antibiotic resistance genes in rectal swabs corresponding with extraintestinal isolates[d] | |
| $bla_{CTX-M-1-Family}$ | $-1.26$ (5.49%) $\pm$ 0.35 |
| $bla_{OXA-48}$ | $-1.13$ (7.39%) $\pm$ 0.26 |
| $bla_{VIM}$ | $-1.1$ (7.97%) $\pm$ 0.31 |

[a]This analysis was not possible for five episodes because there was no growth on the selective plates used in the routine screening for the collection of the intestinal isolate.

[b]This analysis was not possible for one episode because $bla_{KPC}$ was not screened for by qPCR in this study, and it was detected in the extraintestinal isolate.

[c]This analysis was only possible for 4 episodes because in the other cases, either the intraintestinal or the extraintestinal isolate was not stored in the hospital's bacterial collection.

[d]"RLs" stand for Relative Loads.

($P < 0.05$). No significant differences were detected for the rest of the genes, despite all of the RLs being either High or Very High when extraintestinal isolates were detected.

**Tracking the relative intestinal loads of resistance genes in comparison with 16SrRNA sequencing and metagenomic sequencing.** *16SrRNA* metagenomic sequencing was performed for all 18 samples collected from the 5-year-old patient P1 (who had the highest number of samples among all of the patients included in the study). Fig. 3 shows the comparison of the RLs of these samples over time against the sequencing results. This patient was included in this study before transplantation and was already colonized by a VIM-producing *K. pneumoniae*. Before transplantation, the effect of the intestinal lavage was obvious from the increase in the $C_t$ values of the *16SrRNA* gene and the fall in the number of reads obtained from the samples (sample points 4 and 5; Fig. 3).

The drop in the total bacterial population in patient P1 paralleled the drop of the RL of $bla_{VIM}$. However, immediately before this drop, the $bla_{CTX-M-1-Family}$, $bla_{OXA-1}$, and $bla_{OXA-48}$ genes appeared with high RLs and then dropped to undetectable levels after the lavage. Then, the total bacterial population quickly recovered (as seen from the decrease in the $C_t$ values of the *16SrRNA* gene and the increase in the number of sequences obtained), with Very High RLs of all 4 tested genes. This recovery was associated with the transient dominance of *Pseudomonas* spp. (at sample point 6) and a consistent codominance of *Klebsiella* spp. and *Pseudomonas* spp. (58.4% to 93.5%) in almost all of the subsequent samples, with the exception of samples 12 and 13. The RLs of all 4 genes remained within an order of magnitude after that time point and coincided with courses of TMX, noncarbapenem $\beta$-lactams, and carbapenems. Additionally, drops in the total bacterial population and in the RL of $bla_{VIM}$ were observed at points 15 and 16 of the graph, in parallel with another course of carbapenems and TMX being given to the patient before a follow-up surgery.

In terms of diversity, the majority of these samples had few species (6 to 32 observed species) and were dominated by *Klebsiella* spp. and/or *Pseudomonas* spp. (50.1% to 93.5%). The exceptions to this were samples 4 and 5, at which no reads were detected, and samples 11, 12, and 13 which coincided with a drop in $bla_{VIM}$ and the point in time that was 12 weeks after the patient received a course of carbapenems and TMX.

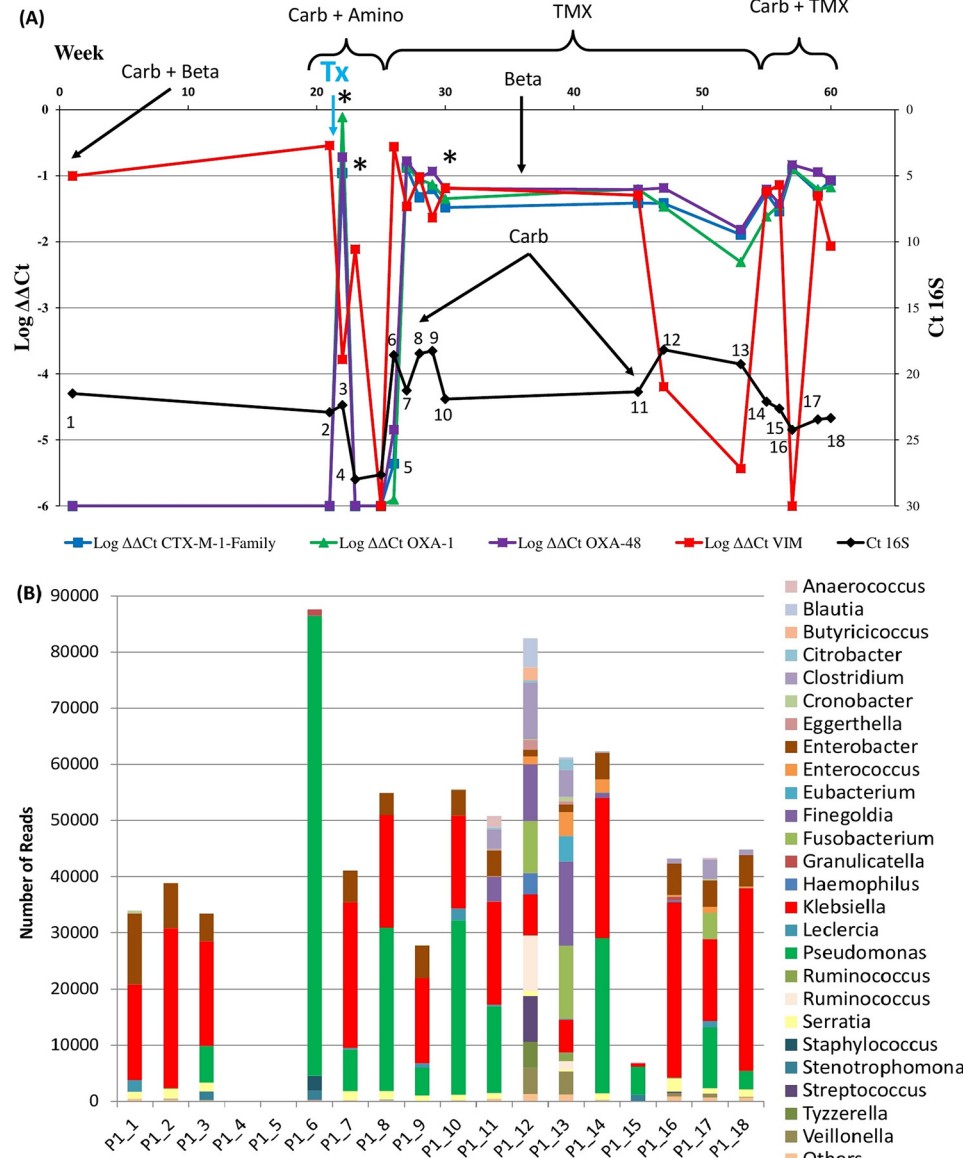

**FIG 3** (A) Relative intestinal loads of $bla_{CTX-M-1-Family}$, $bla_{OXA-1}$, $bla_{OXA-48}$, and $bla_{VIM}$ for Patient P1 combined with data regarding the antibiotics received at the time of sample collection, the extraintestinal isolation of MDROs, and the date of transplantation. The black line represents the raw $C_t$ values for the *16SrRNA* gene and is subject to the scale on the right-hand side of the graph. The numbers on the points of this black line represent the number of samples that correspond to panel B of this figure. (B) *16SrRNA* mass sequencing for the 18 samples obtained from patient P1. The graph represents the genera identified from the consensus of the 9 variable regions of this gene. Carb, carbapenems; Amino, aminoglycosides; TMX, trimethoprim-sulfamethoxazole; Tx, date of transplantation; *, detection of extraintestinal MDROs.

Metagenomic sequencing was performed for samples 1 and 6 directly from the DNA extracted from the swabs. Of the reads obtained from P1_1, 100% of the reads originating from bacteria were classified as Proteobacteria, of which 34% were classified as *K. pneumoniae*, 32% as *Enterobacter cloacae* complex, 14% as *E. coli*, and the rest as other *Enterobacterales*. As for the bacterial reads obtained from P1_6, 46% were classified as *P. aeruginosa*, 35% as *Staphylococcus epidermidis*, and the rest as other Firmicutes and Gammaproteobacteria.

## DISCUSSION

In this study, 169 rectal swabs obtained from 28 liver transplant pediatric patients were analyzed to determine the RLs of the $bla_{CTX-M-1-Family}$, $bla_{OXA-1}$, $bla_{OXA-48}$, and/or

$bla_{VIM}$ antibiotic resistance genes, as well as their time courses and their relation to antibiotic treatments and the extraintestinal spread of MDROs. When detected, the RLs of all 4 genes were well above the typical RL of all *Enterobacterales* (1%) in a healthy gut microbiome. The high RLs of $bla_{CTX-M-1-Family}$, $bla_{OXA-1}$, and $bla_{OXA-48}$ were associated with carbapenem and TMX treatments, in line with the selective pressure that these antibiotics might be exerting on the intestinal bacteria (22). Other studies did indeed show a relationship with intestinal microbiome dysbiosis after antibiotic therapy (16, 23, 24). Perhaps what is even more important is that these treatments sometimes cause a decrease in the RLs of resistance genes, especially after intestinal cleansing, but this is quickly followed by a rapid expansion and dominance of the gut microbiome by MDROs. Occult colonization by carbapenemase-producing organisms has been demonstrated in a mouse model, in which a quick outgrowth in response to antibiotic exposure after previously being below detectable levels was shown (17). This goes in line with our results and highlights the utility of using methods that are able to detect this occult colonization and indicate a need for preventive measures (such as modifications of antibiotic therapies) before allowing MDROs to dominate the intestinal microbiome and possibly spread outside the intestine and cause infections. This is especially important because routine screening methods can easily miss this colonization and might not be able to detect it until the MDROs are already dominating this microbiome. On the other hand, an association was not detected between the presence of $bla_{VIM}$ and the consumption of any particular antibiotic. One reason might be that the presence of this gene in a cassette containing several other antibiotic resistance genes allows for the selection of bacteria harboring this cassette through a wide range of antimicrobial agents (13).

An inverse relationship was observed between the RLs of $bla_{CTX-M-1-Family}$, $bla_{OXA-1}$, and $bla_{OXA-48}$ and noncarbapenem $\beta$-lactams and aminoglycosides. This apparently paradoxical relationship might be due to factors that decrease the selective advantage of the resistant bacteria over the native nonresistant microbiome. These factors can include the mode of administration, bioavailability, synergy, antagonism, and mode of elimination of the antibiotics (25). One study demonstrated a relationship between the oral administration of antibiotics and the minimal selection of MDROs (26), and another demonstrated a relationship between oral antibiotics and the decolonization of carbapenem-producing *K. pneumoniae* (27). Further exploration of the use of antibiotics that result in minimal intestinal microbiome disruption and that prevent intestinal dominance by MDROs among pediatric transplant patients might be an interesting avenue of research to pursue.

The qPCR used in our study to determine the RLs of genes of antibiotic resistance was found to parallel findings through more complex and time-consuming techniques, such as next-generation sequencing. This was especially evident in patient P1 (Fig. 3) in whom tracking the intestinal dominance via qPCR went in line with microbiome analyses and metagenomic sequencing. Though this was tested for using samples from only one patient, the maintained association throughout the 18 samples shows promise in using this approach to track changes in the gut microbiome. Moreover, as shown by the comparison between the RLs of all of the samples that corresponded with extraintestinal isolates in our cohort, there was an association between the high RLs of organisms harboring genes of antibiotic resistance and the probability that they escape their niches, colonize other sites, and cause infections. In addition, they were able to do so with percentages of intestinal dominance that were much lower than those reported in other studies (14, 28–31).

One limitation of our study is the scarcity of samples from the pretransplantation period to be compared to those of the peritransplantation and posttransplantation periods, a factor that should be addressed in future studies. This is especially important because these patients could have had varied therapies based on their particular diagnoses and different amounts of time from diagnosis to surgery. Nevertheless, our data show the potential of using RLs as biomarkers for evaluating the risk of extraintestinal

**TABLE 4** Primers used for the relative quantification of the intestinal load of the *16SrRNA* gene, *bla*$_{CTX-M-1-Family}$, *bla*$_{OXA-1}$, *bla*$_{OXA-48}$, and *bla*$_{VIM}$

| Target | Name | Sequence (5′ → 3′) | Reference |
|---|---|---|---|
| *16SrRNA* gene | P891F | TGGAGCATGTGGTTTAATTCGA | 36 |
| | P1033R | TGCGGGACTTAACCCAACA | |
| *bla*$_{CTX-M-1-Family}$ | CTXM1F | AAACCGGCAGCGGTGGC | This study |
| | CTXM1R | ACGGCTTTCTGCCTTAGGTTG | |
| *bla*$_{OXA-1}$ | OXA1F | CTGTTGTTTGGGTTTCGC | This study |
| | OXA1R | GCTACTTTCGAGCCATGC | |
| *bla*$_{OXA-48}$ | OXA48F | TTGGTGGCATCGATTATCGG | 37 |
| | OXA48R | GAGCACTTCTTTTGTGATGGC | |
| *bla*$_{VIM}$ | VIMF | GTTTGGTCGCATATCGCAACGC | This study |
| | VIMR | AAGCAACTCATCACCATCACGG | |

spread of MDROs, monitoring the RLs in relation with different antibiotics given, and implementing preventive infection control methods among pediatric transplant patients. Additional studies targeting different cohorts can shed light on the possibility for using this approach on other types of patients, as well.

## MATERIALS AND METHODS

**Study design.** The study was performed at the Hospital Universitario La Paz (HULP) in Madrid, Spain. HULP is a tertiary care center with 1,200 beds and is the only Spanish center that offers all kinds of pediatric solid organ transplants. The patients included in this retrospective study were all those that were under the care of the Pediatric Liver Transplant Program from April 2018 until December 2019. Pediatric liver transplant patients or transplant candidates who were included in this study during that period were selected for inclusion based on the first sample received during this time frame, and they were followed for the remainder of the study period. The patients were numbered in chronological order, based on their recruitment, from P1 to P28. During the study, there was an ongoing outbreak of OXA-48 and VIM-producing *Klebsiella pneumoniae* at HULP (32). Therefore, rectal swabs were routinely collected from these patients for the epidemiological surveillance of MDROs and were reused in this study. The samples were labeled with the patient code and were numbered by collection order (i.e., P1_1, P1_2, and so on). All of the samples received from the patients included in this study were analyzed, irrespective of whether the patient was during the preoperative, operative, or postoperative period, and of the ward from which the samples were received. The patients were then grouped into two groups. The group 1 patients are those from whom the first sample was received immediately after transplant surgery and who were inpatients upon inclusion. The group 2 patients are those from whom the first sample was received during the follow-up period and who were outpatients upon inclusion.

Upon receipt, the rectal swabs were cultured on MacConkey plates containing 4 $\mu$g/mL cefotaxime (custom made by Maim, Madrid, Spain) for the selection of ESBL-producing organisms and on ChromID CARBA SMART agar plates (bioMérieux, Marcy l'étoile, France) for the detection of carbapenemase and/or OXA-48 producing organisms. The rectal swabs were then suspended by vigorous shaking in 0.5 mL of TE buffer (10 mM Tris and 1 mM EDTA; pH = 8.0) and stored at −20°C until used (14). When possible, bacterial isolates that were cultured on these media were retrieved from the collection of the Microbiology Department for clonality analyses. Moreover, MDROs that were isolated from clinical samples obtained from other body sites of the patients (i.e., those not from rectal swabs) during the study period and were stored in the bacterial collection were recovered. These isolates are collectively called "extraintestinal isolates" in this study.

**DNA extraction.** Total DNA was extracted from 100 $\mu$L of the rectal swab suspensions that were previously diluted with 900 $\mu$L of TE buffer to avoid inhibition of the quantitative PCR. The mixture was heated at 95°C for 20 min and was then subjected to mechanical lysis at 7,000 rpm for 70 s using MagNA Lyser Green Beads in a MagNa Lyser instrument (Roche, Mannheim, Germany). The MagNa Pure Compact system (Roche, Mannheim, Germany) was then used to extract the total DNA from the samples using the MagNA Pure Compact Nucleic Acid Isolation Kit I (Roche, Mannheim, Germany). The extracted DNA was stored at −20°C until used.

**Primer design and validation.** 2 primer pair sequences were obtained from the literature, and 3 primer pairs were designed in this study. Table 4 shows the primer sequences and their target genes. The primers were designed using whole-genome sequences for *Klebsiella pneumoniae* harboring the targeted $\beta$-lactamase genes. They were then tested on a series of 10 *K. pneumoniae* strains from our bacterial collection that were previously sequenced and known to harbor these genes (13).

The efficiency of the qPCRs was determined by first culturing the *K. pneumoniae* strains harboring the targeted gene(s) on Columbia Agar (bioMérieux, Marcy l'étoile, France). A 0.5 McFarland suspension (equivalent to $10^8$ colony forming units [CFU]/mL) was prepared from the cultured bacteria and was serially diluted (1:10) 6 times. DNA was then extracted from all dilutions as described above. qPCRs (described below) using the primer pairs listed in Table 4 were performed on all of the samples (one qPCR target per tube). The threshold cycle (C$_t$) values obtained were then plotted against the respective CFU/mL values, and the efficiency of the reaction was calculated as $10^{-1/slope} - 1$ (33). To test for the

effect of dilution on the samples, the $C_t$ values of each of the primer pairs targeting the genes of resistance were normalized to the $C_t$ value of the *16SrRNA* gene (expressed as $Log_2^{-\Delta Ct}$) and were plotted against the fold dilution of each tube. The slope of the line resulting from this graph was used to evaluate whether dilution influences the $Log_2^{-\Delta Ct}$ values. All of the validation experiments were performed in triplicate, and positive and negative controls were included in all reactions. The efficiency values of the qPCRs using the primer pairs described in Table 4 ranged from 95.54% to 109.78%, and the dilutions did not have an effect on the $\Delta C_t$ values (slope of the curve was approximately equal to 0).

**Quantification of the relative intestinal load of *bla*<sub>CTX-M-1-Family</sub>, *bla*<sub>OXA-1</sub>, *bla*<sub>OXA-48</sub>, and *bla*<sub>VIM</sub>.** qPCRs targeting the $bla_{CTX-M-1-Family}$, $bla_{OXA-1}$, $bla_{OXA-48}$, $bla_{VIM}$, and the *16SrRNA* gene were performed for each sample using the primer pairs listed in Table 4. All of the genes were simultaneously tested for in independent tubes. The reaction mixture consisted of 10 $\mu$L of 2X PowerUP SYBR Green Master Mix (Applied Biosystems, Waltham, MA, USA), 0.05 $\mu$M each of the respective primer pairs, 6 $\mu$L of $H_2O$, and 2 $\mu$L of extracted DNA. The reactions were run in a CFX Connect Real-Time System (Bio-Rad, Madrid, Spain), using the following conditions: 95°C for 3 min, followed by 40 cycles of 95°C for 15 s and 60°C for 1 min. A melting curve analysis was performed after each run.

The intestinal relative loads (RLs) of the antibiotic resistance genes were determined by normalizing their $C_t$ values to that of the *16SrRNA* gene (which represents the total bacterial population in the sample) and were expressed as $\Delta C_t$ values (34). These values were then normalized to the $\Delta C_t$s obtained from pure bacterial cultures harboring the respective genes in order to express the results in $Log\Delta\Delta C_t$ (20). The $Log\Delta\Delta C_t$ values are on an inverse logarithmic scale on which 0 represents a RL that is equivalent to a pure bacterial culture (i.e., 100% of the bacterial population harbor the resistance gene), –1 represents 10% of the total bacterial population, –2 represents 1%, and so on, until –6, which is our detection limit, representing 0.0001% of the total bacterial population. The percent relative load (%RL) was calculated using the formula $10^{Log\Delta\Delta Ct} \times 100$.

**Clonality analyses.** In the cases in which it was possible to retrieve the intraintestinal and extraintestinal isolates, clonality analyses were performed using Random Amplified Polymorphic DNA (RAPD) (35). The primers OPA-2 (5′-TGCCGAGCTG-3′), OPA-12 (5′-TCGGCGATAG-3′), and OPA-18 (5′-AGGTGA CCGT-3′) were used. The thermal cycling conditions were 94°C for 5 min, followed by 45 cycles of 94°C for 1 min, 37°C for 1 min, 72°C for 2 min, and a final step at 72°C for 2 min. The amplicons were run on 2% agarose gels, and the profiles obtained were compared to determine clonal relatedness.

**Next-generation sequencing.** To compare the data obtained via qPCR to the microbiome composition, all 18 samples obtained from the patient labeled P1 (the first patient included in the study and the longest followed) underwent *16SrRNA* metagenomic sequencing. DNA was extracted from the samples as described above, and the Ion 16S Metagenomics Kit (Thermo Fisher, USA) was used, according to the manufacturer's instructions. Libraries were prepared using the Ion Plus Fragment Library Kit (Thermo Fisher, USA) and Ion Xpress Barcode Adapters (Thermo Fisher, USA), according to the manufacturer's instructions. Purification steps were performed using the Mag-Bind TotalPure NGS (Omega, USA). Libraries were sequenced using the ION Chef and ION Gene Studio S5 (Thermo Fisher, USA), and the reads were analyzed through the ION Reporter Software (Thermo Fisher, USA).

Two samples were chosen for metagenomic analysis because they were dominated by either *Klebsiella* spp. (P1_1; the first sample obtained from patient P1) or *Pseudomonas* spp. (P1_6; the sixth sample obtained from patient P1). DNA was extracted from these samples as previously described. The bacterial DNA was then amplified using the GenomiPhi DNA Amplification Kit (GE Life Sciences, USA) and then fragmented using the M220 Focused-Ultrasonicator (Covaris, USA). Libraries were prepared with the Rapid Sequencing Kit (Oxford Nanopore Technologies, United Kingdom), according to the manufacturer's instructions, and sequenced using MinION with the Flongle Adapter and flow cell (Oxford Nanopore Technologies, United Kingdom). Reads were taxonomically assigned using Centrifuge (version 1.0.3).

**Clinical data.** Clinical data were retrospectively retrieved from the hospital information system database. The information included was: main diagnoses, age upon inclusion, sex, date of liver transplantation, colonization status as determined by the Preventive Medicine Department, and antibiotics received during the study period and up to 30 days before inclusion. Episodes of extraintestinal multidrug-resistant bacterial isolates were also recorded, in addition to whether or not they were causing infections, as determined by the attending physician. The study obtained the approval of the local ethics committee (PI-3428).

**Statistical analyses.** Quantitative statistical analyses were performed in order to determine the significance between the RLs that were grouped into different categories, which include: whether or not there was an extraintestinal detection of MDROs at the time of sample collection, whether or not the patients received antibiotics during the 30 days before sample collection, and whether the samples were obtained during the perioperative or follow-up periods. Qualitative statistical analyses were also performed to determine the associations between categorical variables, such as the antibiotic consumption and having a positive rectal swab for one or more of the tested genes. For the quantitative data, the Kolmogorov-Smirnov (for $n > 50$) and Shapiro-Wilk (for $n < 50$) tests were used to test for normality. Two-sided Student's $t$ tests and analyses of variance (ANOVA) were used to compare the normally distributed data, and the Kruskal-Wallis and Mann-Whitney U tests were used to compare the nonnormally distributed data. For the qualitative data, Chi-square tests were used. All of the statistical analyses were performed using the SPSS program (version 24.0; IBM, Armonk, NY, USA), and $P$ values of $<0.05$ were interpreted as being indicative of a statistically significant result.

**Ethical approval.** Ethical approval for the study was obtained by the local ethics committee of Hospital Universitario La Paz (PI-3428). Informed consent was not deemed necessary because the samples included in this study were reused after collection for routine epidemiological surveillance.

**Data availability.** All of the data generated and analyzed during this study are included in this article and in its supplemental material files. The obtained sequences were deposited in GenBank under the Bioproject accession number PRJNA839263. Further inquiries can be directed to the corresponding author.

## SUPPLEMENTAL MATERIAL

Supplemental material is available online only.
**SUPPLEMENTAL FILE 1**, PDF file, 0.3 MB.

## ACKNOWLEDGMENTS

This work was supported by Fondo de Investigaciones Sanitarias, Instituto de Salud Carlos III, grants PI16/01209 and PI19/01356, to J.M. This work was cofinanced by the European Development Regional Fund, "A way to achieve Europe". E.D. received funding from the European Union's Horizon 2020 research and innovation program under the Marie Skłodowska-Curie Individual Fellowship, grant agreement number 796084 and from the Agencia Estatal de Investigación of the Spanish Ministry of Science and Innovation in the form of a Juan de la Cierva – Incorporación grant (number IJC2019-038832-I). The funders had no role in the study design, data collection and analysis, decision to publish, or preparation of the manuscript.

The authors have no conflict of interests to declare.

E.D. performed the lab work described in this article and drafted the manuscript. L.F.-T., M.A.-D., E.C.-B., C.S., and L.E.-G. tracked the patients involved in the outbreak, collected the clinical data, and participated in the acquisition of the samples. G.R.-C. identified the rectal swabs included in the study and performed the bacterial cultures. F.L.-P. performed part of the primer validation experiments and clonality analyses. M.C.-M. performed data analyses and recovered part of the data from the hospital's information system database. S.J.-R. performed part of the PCRs. E.D. and J.M. designed the project. L.H.-L. and J.M. supervised the project. All of the authors participated in the data analyses, review of the manuscript, and drafting of its final version. All authors approved the version currently being submitted.

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
