## [Reviewer comments · Microbiology Spectrum]

Microbiology Spectrum

Intestinal Dominance by Multi-Drug Resistant Bacteria in Pediatric Liver Transplant Patients

Elias Dahdouh, Lorena Fernández Tomé, Emilio Cendejas-Bueno, Guillermo Ruiz-Carrascoso, Cristina Schüffelmann, María Alos Diez, Fernando Lázaro-Perona, Mercedes Castro Martínez, Luis Escosa-García, Sonia Jiménez Díaz, Loreto Hierro Llanillo, and Jesús Mingorance

Corresponding Author(s): Elias Dahdouh, Hospital Universitario La Paz

Review Timeline:

Submission Date:	July 26, 2022
Editorial Decision:	September 21, 2022
Revision Received:	September 30, 2022
Accepted:	October 18, 2022

Editor: Cezar Khursigara

Reviewer(s): Disclosure of reviewer identity is with reference to reviewer comments included in decision letter(s). The following individuals involved in review of your submission have agreed to reveal their identity: Shahrbanoo keshavarz azizi raftar (Reviewer #2)

Transaction Report:

DOI: <https://doi.org/10.1128/spectrum.02842-22>

September 21, 2022

Dr. Elias Dahdouh
Hospital Universitario La Paz
Molecular Microbiology
IdiPaz, Paseo de la Castellana, 261
Madrid 28046
Spain

Re: Spectrum02842-22 (Intestinal Dominance by Multi-Drug Resistant Bacteria in Pediatric Liver Transplant Patients)

Dear Dr. Elias Dahdouh:

Link Not Available

Sincerely,

Cezar Khursigara

Journals Department
Reviewer comments:

Reviewer #1 (Comments for the Author):

This is an interesting study of 28 pediatric liver transplant patients who underwent rectal swabs for epidemiological MDR bacteria evaluation. The report highlights some interesting findings, including a high relative load of several MDR genes overall, association with particular antibiotic classes, and association with "extra intestinal" infections. The discussion is well written. I believe there are several ways the manuscript can be improved.

Major feedback:

1. My primary concern about the manuscript is that a lot of granular data is presented without enough refinement and

interpretation for the reader. While very interesting findings are buried in the text, every sub-section of results contains too much detail obscuring main points. There are 16 pages of results (not including supplementary materials), which is quite long for this study. Many tables include an exhaustive breakdown of each patient, when more of a summary presentation should be presented. By including this degree of detail, the authors are leaving too much of the interpretation work to the reader, making it challenging to complete reading and draw conclusions. I think several of the tables and figures could be transformed into violin plots (or similar) with an axis being time (before vs after a specific antibiotic, or transplant, etc). I also think the text should be decreased by at least 25%, evenly throughout the results section.

2. The authors conclude their introduction with "our aim is to evaluate the potential benefit of tracking RLs of these genes in the gut microbiome as a biomarker for personalized medicine approaches in managing pediatric transplant patients", but that is not what the manuscript accomplishes. The manuscript is more of a descriptive piece documenting the incidence and natural history of MDR genes. Perhaps the authors mean that this aim is the next step in this work, to evaluate how these gene RLs could be used for managing pediatric transplant patients.

3. Important details are missing from methods section 2.1. Where these patients inpatients every time they were sampled? Why were some only sampled a few times but others sampled many times? It is also not clear until much later in the manuscript (and still even not entirely clear) what "extra-intestinal isolates" are. Also, later in the manuscript group 1 and group 2 appear without explanation. Section 2.1 should better explain how patients were selected for this cohort, how they were sampled, how long they were followed, and how they were grouped.

4. The statistical analysis section is vague and does not orient the reader to what the primary analysis is going to be. It foreshadows a very descriptive manuscript.

5. While not possible to change, a major limitation of this study is the lack of pre-transplant data points. If one of the conclusions is that antibiotics leads to increased abundance of MDROs, how can we conclude that without data points prior to the initial antibiotic load (i.e. transplant)?

Minor feedback:

- Is "dominance" the right term? Catchy and powerful, but what is the definition of this?
- Does the presence of these genes equate to drug resistance? In other words, does the presence of these genes ensure the presence of organisms with drug resistance?
- In 2.6 it was not entirely clear to me what "P1" was until I later saw a table listing every subject as P#.
- Line 195, I suspect authors mean sex, not gender.
- I am curious if the authors have any data on how high RLs or newly "positive" RLs related to the overall outbreak at the hospital, interactions with certain staff members, etc.
- Colangitis should be "cholangitis".

Reviewer #2 (Comments for the Author):

09/12/2022

ASM Microbiology Spectrum 02842-22: Intestinal Dominance by Multi-Drug Resistant Bacteria in Pediatric Liver Transplant Patients.

This study focused on the relationship between intestinal loads of antibiotic resistance genes specifically beta-lactamase among pediatric liver transplantation patients. The concept of the study is interesting; the study is well designed and written in good English. Introduction explained the importance and aim of study clearly. The material and method and result sections are well being explained. However, some points need to be clarified for me.

Why the authors used 18 samples from one patient? Are the results obtained from only one patient accurate? Was this P1 patient breastfeed? If yes, do you checked the mother in terms of antibiotic consumption?

Why didn't the authors use, for example, 5 samples at different time points from more than one patient?

Staff Comments:

Preparing Revision Guidelines

- Point-by-point responses to the issues raised by the reviewers in a file named "Response to Reviewers," NOT IN YOUR COVER LETTER.
- Upload a compare copy of the manuscript (without figures) as a "Marked-Up Manuscript" file.
- Each figure must be uploaded as a separate file, and any multipanel figures must be assembled into one file.

- Manuscript: A .DOC version of the revised manuscript
- Figures: Editable, high-resolution, individual figure files are required at revision, TIFF or EPS files are preferred

Please return the manuscript within 60 days; if you cannot complete the modification within this time period, please contact me. If you do not wish to modify the manuscript and prefer to submit it to another journal, please notify me of your decision immediately so that the manuscript may be formally withdrawn from consideration by Microbiology Spectrum.

Dear Reviewers,

We would like to take this chance to thank you for your constructive criticism of our manuscript and your helpful insight regarding how to improve it. We have addressed all the points raised in your reports and what follows is a point-by-point breakdown of the changes done in the manuscript:

Reviewer #1:

Reviewer: My primary concern about the manuscript is that a lot of granular data is presented without enough refinement and interpretation for the reader. While very interesting findings are buried in the text, every sub-section of results contains too much detail obscuring main points. There are 16 pages of results (not including supplementary materials), which is quite long for this study. Many tables include an exhaustive breakdown of each patient, when more of a summary presentation should be presented. By including this degree of detail, the authors are leaving too much of the interpretation work to the reader, making it challenging to complete reading and draw conclusions. I think several of the tables and figures could be transformed into violin plots (or similar) with an axis being time (before vs after a specific antibiotic, or transplant, etc). I also think the text should be decreased by at least 25%, evenly throughout the results section.

Authors: The results section has been extensively modified. Mainly, the following key changes were made:

- The text has been greatly reduced and only the most important findings were stated in the text. It was not always possible to highlight these changes because a great part of these modifications were deleting the text and adding the information to tables.
- The tables were collapsed into much simpler ones where the summary of the data is presented. The table titles were highlighted to indicate that the entire table has been modified.
- Regarding the violin plots, we made several attempts to transform some of the data into them but since there were very little points in the “before” category (*i.e.* before transplantation or antibiotic consumption, etc...) the graphs were not very representative of the data. We instead summarized most of the data, reduced the level of individual details, and presented them in a more global view of the study.
- Sections 3.2 and 3.3 were merged (and so the numbering of the subsequent sections were changed) and a lot of the text was removed and transformed into tables.
- The information in Table 4 in what is now section 3.2 (previously section 3.3) was found to be redundant and therefore the table was removed.
- In what is now section 3.4 (previously 3.5), Tables 5 and 6 were removed and added to Supplementary Tables 1 and 2 since the important and significant associations were mentioned in the text and the tables seemed redundant.
- In section 3.6 (previously 3.7), no significant reduction of the text was made since we believe that the associated Figure is highly complex and would be difficult to understand

without the detailed information given in the text. Various attempts were made and in all of them, the information after reduction seemed lacking.

We hope that by these changes the presentation of the results became clearer and more informative.

Reviewer: The authors conclude their introduction with "our aim is to evaluate the potential benefit of tracking RLs of these genes in the gut microbiome as a biomarker for personalized medicine approaches in managing pediatric transplant patients", but that is not what the manuscript accomplishes. The manuscript is more of a descriptive piece documenting the incidence and natural history of MDR genes. Perhaps the authors mean that this aim is the next step in this work, to evaluate how these gene RLs could be used for managing pediatric transplant patients.

Authors: The authors agree with this point and therefore this sentence was removed from the introduction since it is an "overreaching aim" for potential future applications.

Reviewer: Important details are missing from methods section 2.1. Where these patients inpatients every time they were sampled? Why were some only sampled a few times but others sampled many times? It is also not clear until much later in the manuscript (and still even not entirely clear) what "extra-intestinal isolates" are. Also, later in the manuscript group 1 and group 2 appear without explanation. Section 2.1 should better explain how patients were selected for this cohort, how they were sampled, how long they were followed, and how they were grouped.

Authors: Methods Section 2.1 was updated in order to clarify how the patients were selected, how the samples were obtained, how long the patients were followed, how they are grouped, and the nomenclature used. The term "extra-intestinal isolates" was also defined in this section. All the changes are highlighted in the marked-up manuscript. Moreover, Table 2 in the Results Section 3.1 was heavily modified in order to give a better global image regarding the patients and samples included in this study (including an update regarding from which wards the samples were received) and will hopefully answer all the questions raised in this point.

Reviewer: The statistical analysis section is vague and does not orient the reader to what the primary analysis is going to be. It foreshadows a very descriptive manuscript.

Authors: The Statistical Analyses Section 2.8 was updated with more details in order to orient the reader to the types of analyses performed (highlighted).

Reviewer: While not possible to change, a major limitation of this study is the lack of pre-transplant data points. If one of the conclusions is that antibiotics leads to increased

abundance of MDROs, how can we conclude that without data points prior to the initial antibiotic load (i.e. transplant)?

Authors: The authors agree with this point and have added the limitation of the study to the discussion section (highlighted).

Reviewer: Is "dominance" the right term? Catchy and powerful, but what is the definition of this?

Authors: The term "dominance" has been used in several articles (DOIs: 10.1128/mSphere.00450-20, 10.1093/cid/cis580, 10.1186/s12859-019-3073-1, and 10.1038/s41396-022-01287-8, 10.3390/microorganisms9112271, among others). That is why we chose to use this word. However, the authors agree that this word might not be as informative as we thought since there is no consensus about the definition, therefore a definition of the term was added the first time it was used in the introduction (highlighted). The phrase added is "defined in this work as an increase in the abundance of these organisms to over 10% of the intestinal microbiome". The 10% value was chosen since it is 10 times higher than the total population of *Enterobacteriaceae* that are present in a healthy gut microbiome (DOI: 10.1186/s40168-017-0244-z).

Reviewer: Does the presence of these genes equate to drug resistance? In other words, does the presence of these genes ensure the presence of organisms with drug resistance?

Authors: We agree that the degree of expression of the genes may be variable, but it is usually strong and constant for VIM and CTX-M, but may vary from subtle to strong in OXA-48. Given the high turnover of the intestinal contents, and the high amounts of resistance genes found, we think that the presence of the genes implies the presence of organisms with some degree of drug resistance. Moreover, in our dataset, all the resistant isolates that were detected on the differential culture media had their corresponding rectal swab positive for the relative gene. Regarding those that were only positive by PCR, another study design would be needed in order to determine whether this discrepancy is due to the limitations of the culture media used, or the lack of expression of these genes (and therefore the lack of phenotypic antibiotic resistance). Addressing this question, though of great interest, was beyond the scope of the study, especially since this study was retrospective in nature and we were not able to handle the original samples ourselves, but rather we re-used them after they were used for routine cultures.

Reviewer: In 2.6 it was not entirely clear to me what "P1" was until I later saw a table listing every subject as P#.

Authors: Clarification regarding the labeling of patients as P# was added in 2.6. Similar clarifications were added to samples P1_1 and P1_6 (both clarifications highlighted).

Reviewer: Line 195, I suspect authors mean sex, not gender.

Authors: The reviewer is correct. Gender was changed to Sex (also in Table 2).

Reviewer: I am curious if the authors have any data on how high RLs or newly "positive" RLs related to the overall outbreak at the hospital, interactions with certain staff members, etc.

Authors: The authors agree that this would be an interesting piece of data to add to the manuscript. However, this information is not readily available to us and felt as beyond the scope of the study, and therefore information was not sought after. Nevertheless, in a previous study performed in our group targeting adult patients, the first samples collected had high RLs compared to the abundance of *Enterobacteriaceae* in a healthy gut microbiome, but these RLs were lower as compared to the last sample obtained from these patient, especially after having received various antibiotic therapies (DOI: 10.1016/j.cmi.2020.09.054).

Reviewer: Colangitis should be "cholangitis".

Authors: Thank you for pointing out this error. The change has been made.

Reviewer #2:

Reviewer: Why the authors used 18 samples from one patient?

Authors: The authors chose this patient since he was the most followed patient (first patient included in the study). The questions we wanted to explore with this experiment were how the different interventions affect the intestinal microbiome of the patient over time, how intestinal microbiome disruption relates to the relative loads determined by qPCR, and how this disruption is related to extra-intestinal spread of MDROs. Considering this, the authors thought it adequate to perform this experiment on the patient from whom the most number of samples were collected and that was followed for the most period of time during the study. Clarifications regarding these points were added in sections 2.1, 2.6, and 3.7 (highlighted).

Reviewer: Are the results obtained from only one patient accurate?

Authors: The authors agree that general conclusions cannot be reached with one patient, and would like to point out that the conclusions of the study were based on the results obtained from the entire cohort. The data for this patient in particular were generated with the intention of determining the variations within the same patient's intestinal microbiome, and were meant to demonstrate the ability to track these changes using qPCR. Moreover, the maintained associations all throughout the 18 samples showed promise in using this approach, despite not being able to reach a global conclusion. This was mentioned in the discussion of the marked up version of the manuscript (highlighted). Furthermore, the authors would like to mention that a study is under way in our lab that targets a different cohort and aims at further testing the relationship between *16sRNA* metagenomic sequencing and the qPCR results, which might lead to a more global conclusion regarding this aspect.

Reviewer: Was this P1 patient breastfed? If yes, do you checked the mother in terms of antibiotic consumption?

Authors: Patient P1 was included in the study at 5 years of age, and therefore he was not breastfed. The age of the patient was added in section 3.7 (highlighted).

Reviewer: Why didn't the authors use, for example, 5 samples at different time points from more than one patient?

Authors: The reason why we were not able to do so was because the patients were included at different time points, had highly variable interventions, were followed for variable amounts of time, and did not have samples collected at regular time intervals (the study was retrospective and we could only use the samples that were collected by the epidemiology department that followed different criteria at different time points based on the evolution of the outbreak). Because of those uncontrollable conditions, all the samples obtained from the patients were included since there was no clear criteria that we can use and that applies to all the patients so that can “normalize” the samples and choose a number of them for inclusion. Extensive clarifications regarding the patient inclusion and sampling was done in Section 2.1 of the marked up version of the manuscript (highlighted).

Finally, as per the editor’s instructions, please note that the tables and figure legends have been moved to after the references section, and that the figures have been uploaded as separate files. The authors would once more like to thank the reviewers for taking the time to review this manuscript and hope that the changes done have made the manuscript clearer and easier to follow.

October 18, 2022

Dr. Elias Dahdouh
Hospital Universitario La Paz
Molecular Microbiology
IdiPaz, Paseo de la Castellana, 261
Madrid 28046
Spain

Re: Spectrum02842-22R1 (Intestinal Dominance by Multi-Drug Resistant Bacteria in Pediatric Liver Transplant Patients)

Dear Dr. Elias Dahdouh:

Your manuscript has been accepted, and I am forwarding it to the ASM Journals Department for publication. You will be notified when your proofs are ready to be viewed.

Sincerely,

Cezar Khursigara
Editor, Microbiology Spectrum
